# Pre- and Post-Vaccination Measles Antibody and Persistence Up to 5 Years of Age Among Early ART-Treated HIV-Infected, HIV-Exposed Uninfected and HIV-Unexposed Children in Cameroon

**DOI:** 10.3390/vaccines13060584

**Published:** 2025-05-30

**Authors:** Mathurin Cyrille Tejiokem, Emilie Desselas, Thierry Joel Noumsi, Francis Ateba Ndongo, Suzie Tetang Ndiang, Mireille Arlette Fossi, Georgette Guemkam, Berenice Zangue Kenfack Tekougang, Paul Alain Tagnouokam-Ngoupo, Ida Calixte Penda, Albert Faye, Josiane Warszawski

**Affiliations:** 1Centre Pasteur du Cameroun, Yaoundé P.O. Box 1274, Cameroon; emilie.desselas@ird.fr (E.D.); noumsit@yahoo.fr (T.J.N.); berenicenicy25@gmail.com (B.Z.K.T.); tagnouokam@pasteur-yaounde.org (P.A.T.-N.); 2CESP, INSERM, UVSQ, Université Paris-Saclay, 94807 Villejuif, France; josiane.warszawski@inserm.fr; 3Institut de Recherche pour le Développement, 34394 Montpellier, France; 4Centre Mère et Enfant de la Fondation Chantal Biya, Yaoundé P.O. Box 1364, Cameroon; atebfranc@gmail.com (F.A.N.); guemgeorg@hotmail.com (G.G.); 5Centre Hospitalier d’Essos, Yaoundé P.O. Box 5777, Cameroon; ndiangsuzie@yahoo.fr; 6Hôpital Laquintinie de Douala, Douala P.O. Box 4035, Cameroon; fossimireillearlette@gmail.com; 7Département des Sciences cliniques, Faculté de Médecine et des Sciences Pharmaceutiques, Université de Douala, Douala P.O. Box 2701, Cameroon; calix.penda@gmail.com; 8Service de Pédiatrie Générale, Hôpital Universitaire Robert Debré, Assistance Publique des Hôpitaux de Paris, 75019 Paris, France; albert.faye@aphp.fr; 9UMR 1123 ECEVE INSERM, Université Paris Cité, 75010 Paris, France; 10Service de Santé Publique et d’Épidémiologie, Hôpital de Bicêtre, Assistance Publique des Hôpitaux de Paris, 94270 Le Kremlin Bicêtre, France

**Keywords:** measles humoral immunity, persistence of measles antibodies, pre-measles vaccine antibodies, early treated HIV-infected, HIV-exposed uninfected, HIV-unexposed uninfected, low- and middle-income countries, real life conditions

## Abstract

Background/Objective: Variations in measles vaccine antibody response by age and HIV status have been reported. This study assessed measles pre-vaccination status and compared humoral response durability over the first five years of life among HIV-infected (HI) children on early treatment, HIV-exposed uninfected (HEU), and HIV-unexposed uninfected (HUU) children within the ANRS 12225—Pediacam III cohort in Cameroon. Methods: Measles vaccine (MCV) was administered at 6 and 9 months for HIV-exposed infants and at 9 months for HIV-unexposed infants, followed by a measles-mumps-rubella (MMR) dose at 15 months for all. Measles antibody titers were measured pre-vaccination, 1–6 months post-MCV doses, and annually until age 5 using ELISA (Enzygnost, Dade Behring). Results: A total of 496 children were included: 143 HI (median age at cART initiation: 4.2 months, (IQR: 3.2–5.6)), 180 HEU, and 173 HUU. Of these, 456 children were tested pre-vaccination (median age: 6.1 months, IQR: 5.6–6.8), with 6.1% (95% CI: 4.1–8.6) seropositive to measles antibodies, with differences across groups. At 18.4 months (IQR: 18.1–19.9), seropositivity rates were 96.7% (59/61) in HI, 96.8% (90/93) in HEU, and 100% (111/111) in HUU groups. For children following the 6 + 9 + 15-month or 9 + 15-month MCV schedules, seropositivity at 18, 36, 48, and 60 months was 96%, 89%, 87%, and 88%, respectively, with no significant differences between groups. Conclusions: Early cART initiation in HI children may result in a robust initial measles antibody response, with comparable persistence of antibody titers across all groups up to five years.

## 1. Introduction

Measles remains one of the leading threats to young children living in low-income countries (LICs) despite the availability of safe and cost-effective vaccines [1,2]. In 2023, there were 128,000 measles-related deaths globally [3], particularly in LICs and regions with weak health infrastructures, including sub-Saharan Africa, where almost 90% of 1.4 million children and adolescents living with HIV reside [4]. The main cause of measles-related mortality is the difficulty in maintaining high measles vaccine coverage to interrupt human-to-human viral transmission [5,6,7]. Like other nations, Cameroon continues to experience measles outbreaks despite the implementation of the Expanded Program on Immunization (EPI), launched in 1976 [8]. The official coverage rate of the first measles dose administered at 9 months varied between 74% to 71% from 2001 to 2023. The second dose of the measles vaccine, targeting children aged 15 months, was introduced in 2020, with an estimated uptake of 45% in 2023. In the same year, the country reported 6105 measles cases, an increase of approximately 90% compared to 2022 [9]. According to the most recent Demographic and Health Survey (DHS) conducted in 2018 in Cameroon, HIV prevalence among adults aged 15–49 years declined from 5.4% in 2004 to 4.3% in 2011, and further to 2.7% in 2018. The HIV prevalence was higher among women (3.4%) than men (1.9%) [10]. Nationally representative data on HIV prevalence in children under 15 are lacking, as this group is excluded from DHS. In 2023, an estimated 27,960 children under 15 were living with HIV (5.7% of all people living with HIV), with 3657 new pediatric infections reported. Among 12,957 HIV-exposed infants, 96.8% received early PCR testing, with a 3.3% positivity rate. The overall mother-to-child transmission rate, including during breastfeeding, was estimated at 15% at 12 months [11]

It is well documented that HIV infection impairs humoral and cellular immune responses consisting of lower antibody titers, more rapid antibody declines over time and a lack of recall responses compared with HIV-uninfected children [12,13,14]. Early initiation of combination antiretroviral therapy (cART), either before immunological deterioration or prior to vaccination, may help enhance the immune response to vaccines [15,16]. Recent evidence also indicates that HIV-exposed uninfected children may have an increased risk of mortality and morbidity compared with HIV-unexposed uninfected onces [17,18,19]. The reasons are still not well understood, but it is postulated that, uninfected children exposed to HIV particles or cART during pregnancy may undergo perturbation of their immune system with consequences on lymphocyte differentiation and functional capabilities [20,21]. Several studies reported the impact of this dysfunction observing low attenuated vaccine-specific antibody levels in HIV-exposed uninfected (HEU) compared with HIV-unexposed uninfected (HUU) children [22,23,24]. Many studies have focused on measles vaccine responses in either short-term or cross-sectional analyses [14,25].

To address some of these gaps, we analyzed data from the ANRS 12225—Pediacam III cohort, which included early-treated HIV-infected children (HI) followed from birth or diagnosis (before 7 months of age), alongside two uninfected control groups: HIV-exposed uninfected (HEU) born to HIV-infected mothers and HIV-unexposed uninfected (HUU) children. Designed to reflect near-real-life conditions with structured follow-up, this study aimed to evaluate measles pre-vaccination status, immunogenicity, and the persistence of measles antibody responses across these three groups. Additionally, we compared the humoral response to the measles vaccine (MV) at different time points during the first five years of life in these groups.

## 2. Materials and Methods

### 2.1. Data Source: The ANRS 12140/12225—Pediacam Study

The Pediacam study is coordinated by the Centre Pasteur of Cameroon (CPC) and is ongoing in three referral hospitals in Cameroon: the Mother and Child Center of the Chantal Biya Foundation (MCH/MCC-CBF) and Essos Hospital Center (EHC), both in Yaounde and the Laquintinie Hospital in Douala (LH). It was originally designed to evaluate, among other objectives, the humoral response of early-treated HIV-infected children to vaccines administered under the Expanded Program on Immunization (EPI).

Briefly, from November 2007 to October 2011, three groups of children were constituted as previously described [26,27] for prolonged follow-up: early-treated HIV-infected children (HI) followed from birth (n = 69) or at diagnosis before 7 months of age (n = 141), HIV-uninfected born to HIV-infected mothers (HEU, n = 205), and HIV-uninfected born to uninfected mothers (HUU, n = 196). Systematic cART was offered to all HIV-infected infants (lopinavir/ritonavir or nevirapine-based therapy; median age: 4 months). All children were followed up and received childhood vaccines (BCG, polio, diphtheria, pertussis, tetanus, hepatitis B, Haemophilus influenza b, measles and yellow fever) according to the national Expanded Programme on Immunization (EPI) calendar. Blood samples were collected from HIV-infected children every three months from inclusion to two years, and every six months thereafter; and for HIV-uninfected children, samples were collected every six months from inclusion to five years. Samples were transported to the Centre Pasteur of Cameroon in Yaounde, where they were processed for care purposes, and the remaining sera were stored at −80 °C. Data were collected at each visit by interviewing the parents/caregivers. Incentives and reimbursement of transport costs were provided to parents/caregivers by the project.

### 2.2. Measles Vaccine Schedule at the Time of the ANRS 12225—Pediacam III Cohort Study

The measles-containing vaccine (SII, 0.5 ml, Intramuscular use) was provided free of charge through routine vaccination services from 2008 to 2012 under the EPI. It was scheduled for this study at 6 months (MCV0) and 9 months (MCV1) for HIV-exposed children (infected-HI or uninfected-HEU), and at 9 months (MCV1) for HIV-unexposed uninfected infants (HUU). Due to the risk of associated diseases from live attenuated vaccines, the national program recommends not administering the measles vaccine if the HIV-infected child is symptomatic or has a CD4 count of <15%. A catch-up dose of the measles vaccine, administered as the Measles-Mumps-Rubella (MMR, Sanofi Pasteur, Marcy-l’Etoile, France), was scheduled at 15 months (MCV2). Given that this dose was currently available at a cost, the project has provided it to the children.

### 2.3. Study Participants

Of the 611 infants included in the ANRS 12225—Pediacam III cohort for prolonged follow-up, 55 (9.0%) were lost to follow-up (LTFU) or died before reaching the age for measles vaccination. Of the remaining infants, 26 (4.3%) who reached the age for vaccination did not receive any MCV (25 HI and 1 HEU). In total, 530 (86.7%) children received at least one dose of MCV. Among these, 5 children developed measles (confirmed wild-type measles infection with the presence of MV-specific IgM antibodies or suspected measles infection before vaccination, 4 HI and 1 HEU), and 456 children had stored samples collected prior to measles vaccination.

Only children who received one or two doses of measles-containing vaccine (MCV) before one year of age, as scheduled and within the following window period, were included in our analysis: MCV0 administered at 6 months [4.5–8.2] and MCV1 administered at 9 months [8.2–12.0]. The MMR vaccine was considered if administered within the period [12.0–20.0] months. Additionally, only measles serological results obtained within 1 to 6 months after measles vaccination were considered. Based on these criteria, 11 children who received MCV after one year of age, 9 children who were vaccinated but tested outside the 1- to 6-month interval, and 6 children who received only the MMR vaccine were excluded from this analysis, resulting in a total of 496 children included (HI (n = 143), HEU (n = 180), and HUU (n = 173)) (Figure 1).

### 2.4. Main Outcome and Principal Exposure Definitions

The main outcome studied was measles antibody titers, which were measured in batches at CPC by technicians blinded to children’s HIV status and personal information. All samples collected before and after MCV were tested using a commercial ELISA kit according to manufacturer’s instructions (Enzygnost Antimeasles virus/IgG, Dade Behring, Germany). The assay was calibrated using an International Reference reagent for the measles vaccine (National Institute for Biological Standards and Control (NIBSC) #92/648, Hertfordshire, UK). The results were classified according to cut-off values defined by the manufacturer: negative (ΔOD < 100 mUI/mL), equivocal (100 ≤ ΔOD ≤ 335 mUI/mL), or positive (ΔOD > 335 mUI/mL), with the latter assumed to represent protective levels of antimeasles virus antibody. Participants with a titer of 1 were assigned an arbitrary concentration of 2 mUI/mL.

The principal exposures investigated were the children’s HIV status (HI, HEU, and HUU) and the number of measles-containing vaccine (MCV) doses received.

### 2.5. Statistical Analysis

Maternal and child characteristics at inclusion or at the first dose of measles vaccination were compared between the four groups of children using Pearson’s chi-square test or the Kruskal–Wallis rank sum test, as appropriate. We subsequently merged the two HI groups, postulating that after cART initiation (at a median age of 4 months), an improvement of immunological characteristics will be achieved in both groups by around 9 months of age (MCV1). Geometric mean titers (GMTs) among groups were displayed with 95% Bootstrap confidence intervals. Proportions of seropositive children to measles antibodies and GMTs were compared using a chi-square test or analysis of variance (ANOVA), as appropriate, between groups before and after vaccination within the window period of 1 to 6 months during the first 24 months of age. Comparisons were also made for measles antibody persistence at 18 (15–27), 36 (30–42), 48 (42–54), and/or 60 (54–61.5) months of age among children who received MCV twice (9 + 15 months) or thrice (6 + 9 + 15 months). All analyses were performed using R 4.0.5 software. For all tests, a *p*-value < 0.05 was considered statistically significant.

## 3. Results

### 3.1. Study Population

As shown in Figure 1, a total of 496 children (52.6% female) were considered for this analysis, consisting of 143 HIV-infected (followed from the first week of life or from diagnosis <7 months), 180 HEU, and 173 HUU children. The groups differed significantly in terms of the children’s age at inclusion, with a lower proportion of highly educated parents among those of HIV-infected children. However, a similar proportion of children were observed across the groups regarding gender, prematurity, mother’s marital status, and number of children per home. At inclusion, compared with HIV-infected followed from first week of life, HIV-infected infants followed from diagnosis <7 months were more likely to experience wasting (13.7% vs. 35.9%; *p* < 0.01), to be immunosuppressed (mean [SD] CD4%: 28 [12.5] and 20 [12.5]), and to have been previously in clinical stage 3 and 4 (9.8% vs. 41.3%; *p* < 0.001) (Table 1).

### 3.2. Feasibility of the Measles Immunization Schedule

Of the 496 children included, the proportion vaccinated according to the planned study measles immunization schedule was 30.8% (44/143) for HIV-infected, 45.0% (81/180) for HEU, and 83.2% (144/173) for HUU children, resulting in an overall vaccination coverage of 54.2% (95% CI: 49.7–58.7). When accounting for those who received vaccinations at 9 and 15 months among HIV-infected and HEU children, the proportions increased to 65.0% (93/143) and 70.0% (126/180), respectively. The median age at administration of MCV0, MCV1, and MCV2 was 6.2 months (IQR: 6.0–6.3; n = 198), 9.2 months (IQR: 9.0–9.4; n = 450), and 15.3 months (IQR: 15.1–16.1; n = 389), respectively. The median (IQR) post-vaccination sampling periods were 2.3 months (1.0–6.0) for MCV0, 3.0 months (1.0–6.0) for MCV1, and 3.0 months (1.0–6.0) for MCV2, with comparable results across the groups.

### 3.3. Pre-Vaccination Antimeasles Virus Antibody Among Children’s Groups

Globally, 456 children (153 HI, 156 HEU, and 147 HUU) were tested before any measles vaccination at a median age of 6.1 months (IQR: 5.6; 6.7), significantly different between the children’s groups (*p* = 0.03, Table 2). Among them, 6.1% (28/456) (95% CI: 4.1–8.6) were classified as seropositive and 3.9% (18/456) as equivocal to anti-measles virus antibody. Before 6 months of age, the proportions of children seropositive to measles antibodies were statistically different between the HI group and the uninfected groups (HEU and HUU) (1.6% vs. 14.6% vs. 15.0%, *p* = 0.011) (Table 2). Between 6 to 9 months of age, the proportion of children seropositive to antimeasles virus antibody was globally low (4.7%) and not significantly different between groups (*p* = 0.232) (Table 2). Subsequently, between 9 and 12 months, the presence of antimeasles virus antibody before vaccination was still observed in all groups of children (*p* = 0.232), although at low levels.

### 3.4. Humoral Immunogenicity After Measles Vaccination

As shown in Table 3, the proportion of HIV-infected children who were seropositive to measles vaccine antibodies was globally similar to that of HEU and HUU children. This trend was also observed for GMT results. A significant increase in measles antibody seropositivity was noted across all three groups of children, from 78%, 82%, and 76% respectively for HI, HEU, and HUU children after MCV1 at 9 months, to 92%, 97%, and 99% after receiving MCV2, with the highest titer observed in the HUU group.

The HEU and HI groups showed similar proportions of children who were seropositive to measles vaccine antibodies after the 9-month vaccination compared with the 6-month vaccination (75% vs. 82%; *p* = 0.97 and 73% vs. 78%; *p* = 0.999). Across all groups, a positive trend was observed, with an increasing proportion of children seropositive to measles vaccine antibodies according to the number of MCV doses received: 1MCV (6-month), 1MCV (9-month), 2MCV (6 + 9-month), 1MCV + 1MMR, and 2MCV + 1MMR in HEU (*p* < 0.001) and HI (*p* = 0.002); 1MCV (9 months) and 1MCV + 1MMR in HUU (*p* < 0.001). A similar pattern was observed with the geometric mean titer in HEU (*p* < 0.001), HI (*p* < 0.001), and HUU (*p* < 0.001) (Table 3).

Regardless of HIV status, a subgroup analysis of children seropositive for antimeasles virus antibody prior to vaccination revealed that among the 28 children identified, 14 were tested after receiving the 9-month MCV. Of these, 6 were seronegative, and 8 were seropositive, including 4 with GMTs lower than their pre-vaccination level. Of the 6 seronegative children, 5 were subsequently tested after receiving MMR and became seropositive.

### 3.5. Persistence of Antimeasles Antibodies Among Children Up to 60 Months

We analyzed measles antibody results available on average at 18 months as baseline (n = 198), and at 36-, 48-, and/or 60-month visits, restricted to children who received 6 + 9+15- or 9 + 15-month MCV schedule. The proportions of children seropositive to measles antibodies were 96% (95% CI: 93–99%), 89% (95% CI: 85–94%), 87% (95% CI: 82–92%), and 88% (95% CI: 82–93%), respectively with no significant differences between the groups (Figure 2 and Figure 3). Of the 198 children, 154 (77.8%) remained measles seropositive at baseline, 36, 48, and/or 60 months. Two children (1.0%) remained measles seronegative or indeterminate at all visits. Fourteen children (7.1%) who were initially seropositive became seronegative or indeterminate but later reverted to seropositive. Ten children (5.1%) transitioned from seropositive to seronegative or indeterminate, and one child (0.5%) went from seropositive to seronegative, then back to seropositive, and finally seronegative again. Ten children (5.1%) transitioned from seronegative or indeterminate at baseline to seropositive, and seven children (3.5%) went from seronegative or indeterminate at baseline to seropositive.

Regarding the geometric mean titer (GMT) for measles antibodies, the rate of decline varied from the 18-month baseline titer (R = 1). The proportion of children with declining measles antibody levels (0 < R < 1) progressively increased: 58% at 36 months, 74% at 48 months, and 77% at 60 months, with no differences according to HIV status (Figure 4 and Figure 5).

## 4. Discussion

Early HIV screening techniques for children and cART are now available, though access to these interventions may not always be easily sustained in real-life settings. Several studies, have shown that early initiation of cART in HIV-infected children is not only feasible but also enhances their life expectancy and reduces the infectious diseases and nutritional issues linked to HIV-induced immunosuppression [26,28,29]. One key concern is whether HIV-infected children treated early with cART can adequately respond to EPI vaccines, particularly in countries where vaccine-preventable pathogens continue to circulate. Another concern is to confirm that HIV-uninfected children born to HIV-infected mothers do not experience impaired responses to EPI vaccines.

The most important finding of this study is that all groups of children were quantitatively similar of both the proportion and GMT of antimeasles virus antibodies following MCV during the first five years of life. However, HIV-infected children exhibited a slightly lower GMT, except when two doses of MCV were administered at 6 and 9 months, or when two doses of MCV plus MMR were given at 6 and 9 months, followed by a dose at 15 months on average. Furthermore, after the scheduled booster at 15 months, antibody titers increased in all groups, supporting the findings of Siberry et al. and Mutsaerts et al. [30,31], which indicated that the proportion of children with protective antibody levels increases with the number of doses. These results suggest that HI children who receive early cART mount an immune response to MCV similar to that of HEU and HUU children over the first five years of life, although the functionality of the antibodies was not assessed in this study [16].

We also found low proportions of children with seropositive antimeasles antibodies after the 6-month measles-containing vaccine (76%, 73%, and 79% among HEU, HUU, and HI, respectively), indicating children produce antibodies and thus respond to the MCV. Similarly, low rates have been reported after one dose of the measles vaccine from other developing countries [32]. Also, a study on the safety and immunogenicity of early-initiated vaccination in children born to infected mothers prior to highly active antiretroviral therapy (HAART) initiation showed that the vaccine was well tolerated and immunogenic at 6 months in infected children [32,33]. On the other hand, a study conducted in Zambia shows good and similar primary antibody responses after administration of an Edmonston-Zagreb strain vaccine at nine months to HI and HEU children (88% vs. 94%, *p* = 0.3). However, a rapid waning of measles antibodies was observed in HIV-infected children [34]. The response after a single dose of MCV at nine months in HIV-unexposed, uninfected children was lower than the anticipated 85% [35]. This low level could may be due to improper vaccine administration, as vaccination was conducted routinely. However, no evidence of cold chain disruption was found after discussions with EPI personnel. The persistence of maternal antibodies could also be one of the causes, but only 6% were still positive after six months. We can also question the use of the standardized and commercially available ELISA method, which is less sensitive than plaque reduction, the gold standard [36,37]. A study conducted in Switzerland on vaccinated students who tested negative by enzyme immunoassay found positive measles virus-specific antibody levels using immunofluorescence and plaque neutralization tests [38]. The use of serological antibody concentrations via ELISA is easily manageable by clinical laboratories and is considered an acceptable surrogate for MCV response [36].

Despite the support provided through this project, only 54.2% of enrolled children received the measles vaccine as planned, with an even lower proportion among those living with HIV. This highlights challenges in immunizing this population at a time when pediatric access was limited. Contributing factors may include insufficient knowledge among healthcare staff and inadequate caregiver counselling [39]. It is well established that achieving adequate immunization coverage and adherence to the vaccination schedule is essential [40].

The proportion of children with pre-vaccination antimeasles virus antibodies was globally low (6.3%) and significantly lower among HIV-infected children (1.6%) compared with HEU (14.6%) or HUU (15.0%). Similar results were found in a study conducted in Malawi [41]. This low level contrasts with the situation observed in other studies on the persistence of maternal antibodies, which reported about 25% of maternal antibodies among HIV-exposed uninfected children in Asia [42,43,44,45,46]. Furthermore, an observational study conducted in Zambia found that at 6 months of age, before any vaccination, 8.9%, 16.7%, and 42.3% of HI, HEU, and HUU were still seropositive to the measles antibody [47]. Another study reported that at 6 months of age, 4.7% of children had antibodies to 2.7% at 9 months in healthy (HIV-) children [42]. Regardless of HIV status, children born to HIV-infected mothers may lose maternally acquired antibodies more rapidly than those born to HIV-negative mothers [23]. Some evidence suggests that although infected mothers have similar antibody titers, HIV-positive children have fewer antibodies in the cord blood compared with HIV-negative children. This has been associated with HIV infection impairing the transfer of antibodies from mother to child [48,49]. Likewise, lower antibody levels in HI children may result from their mothers primarily acquiring immunity through vaccination, which is less robust than natural immunity from wild-type virus exposure [43,48]. In our study, the majority of mothers were born before the introduction of measles vaccines in Cameroon. In addition to the reduced quantity of antibody transfer, some studies have also questioned the quality of anti-measles antibodies. Reports indicate lower measles neutralizing activity in sera from children under 4 months of age born to HIV-infected mothers compared with matched controls born to HIV-uninfected mothers [50]. Although not specifically evaluated in our study, differences in measles antibody levels between child groups are possibly due to maternal antibody levels, the efficiency of placental transfer (which may be impaired by maternal malaria, HIV infection, or prematurity), and the rate of catabolism after birth [47]. These findings confirm that children living in settings with ongoing measles outbreaks face an increased risk of measles before reaching the age of routine vaccination. This suggests the importance of reinforcing the implementation of National EPI recommendations, which stipulate that children born to HIV-infected mothers should be immunized against measles early (i.e., at both 6 and 9 months). However, implementing this recommendation in our setting remains challenging for several reasons: the HIV status of children is often unknown in routine immunization programs, and under ideal conditions, the CD4 platform for measuring CD4 lymphocytes is not always available to assess whether severely immunocompromised HIV-infected infants should be vaccinated.

In analyzing the kinetics of measles vaccine antibody evolution, all the child groups showed an overall increase in antimeasles virus antibodies following MCV1 and/or MCV2, with a similar trend observed in GMT levels. From the administration of MCV2 until approximately five years of age, a slight decline was noted, likely due to waning immunity. In the context of ongoing measles virus circulation in our country, this may warrant consideration of an additional vaccine dose during this period. Assessing whether antibody levels persist over time is crucial, especially given the measles outbreaks reported during this period. By five years of age, antibody titers were nearly identical across all three groups, suggesting that early initiation of cART in HI children may support a sustained immune response to MCV [31]. Additionally, the significant rise in antibody titers at 36 months compared with 30 months suggests a vaccination campaign, as the increase was observed across all groups. Notably, a nationwide measles vaccination campaign was conducted in November 2015 for children aged 9 to 59 months in Cameroon.

This study has several strengths. Its prospective design, inclusion of two control groups, and follow-up over 60 months of age provide valuable medium-term insights into vaccine-induced immunity in early-treated HIV infection. Conducted in Yaounde and Douala, the two largest cities in Cameroon, the study population reflects a diverse, multiethnic community, making the findings broadly representative of urban and suburban populations. However, this study has some limitations. Some important data were not adequately collected, such as the feeding mode during follow-up. Subclinical infections could not be definitively ruled out, though evidence of asymptomatic antibody boosting was observed.

## 5. Conclusions

This study shows that early cART in HIV-infected children is associated with a robust initial measles antibody response. Additionally, a reassuring antibody response was also observed in HEU. Low pre-vaccination anti-measles virus antibodies particularly in HI strongly suggests the need for adherence to the EPI routine schedule.

## Figures and Tables

**Figure 1 vaccines-13-00584-f001:**
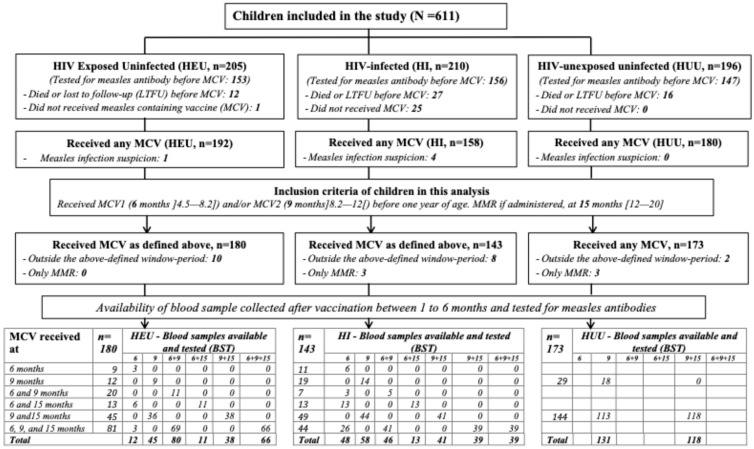
Flow diagram for study participants (LTFU: loss to follow-up; MCV: measles-containing vaccine; MMR: measles, mumps, and rubella).

**Figure 2 vaccines-13-00584-f002:**
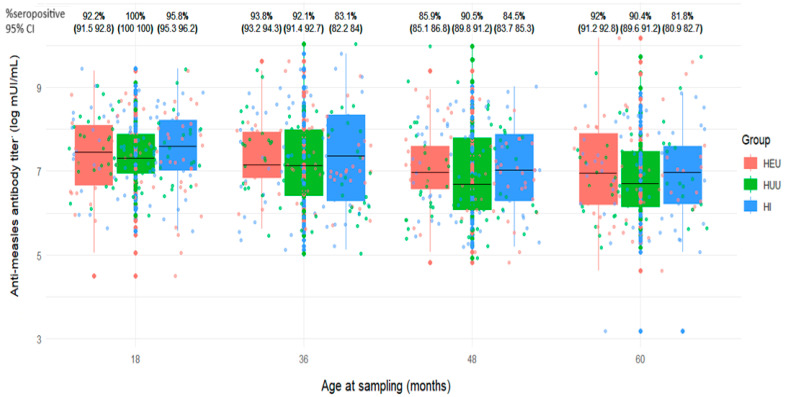
Distribution of anti-measles antibody titers and seropositivity levels at various time points between 18 and 60 months among children who received measles-containing vaccine at either 9 and 15 months or at 6, 9, and 15 months (HEU: HIV-exposed uninfected; HUU: HIV-unexposed uninfected; HI: HIV infected; CI: Confidence interval).

**Figure 3 vaccines-13-00584-f003:**
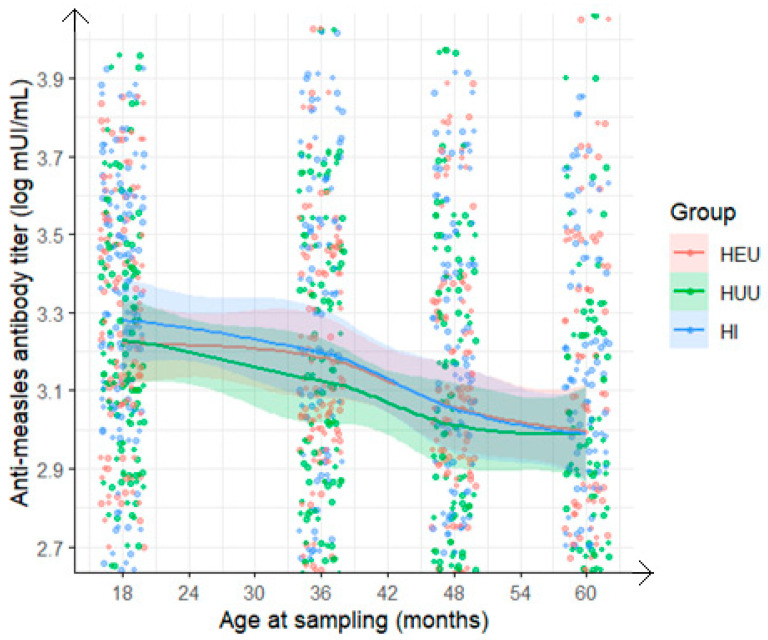
Evolution at various time points from 18 to 60 months of anti-measles antibody titer among children who received measles-containing vaccine at either 9 and 15 months or at 6, 9, and 15 months (HEU: HIV-exposed uninfected; HUU: HIV-unexposed uninfected; HI: HIV infected).

**Figure 4 vaccines-13-00584-f004:**
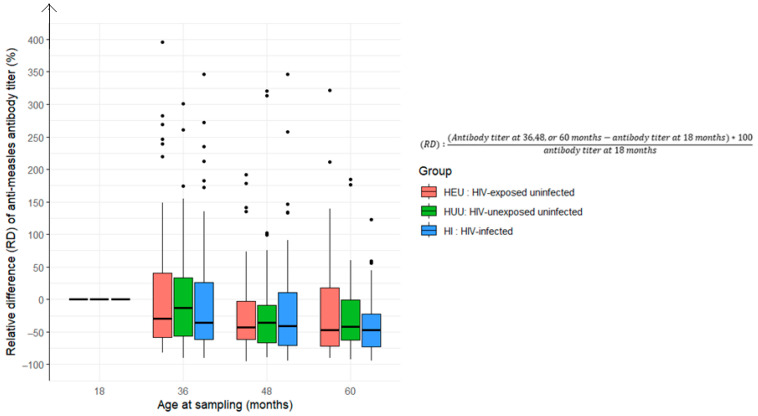
Relative difference in anti-measles antibody titers at 36, 48, and 60 months compared to baseline levels at 18 months among children who received the measles-containing vaccine at either 9 and 15 months or at 6, 9, and 15 months (HEU: HIV-exposed uninfected; HUU: HIV-unexposed uninfected; HI: HIV infected).

**Figure 5 vaccines-13-00584-f005:**
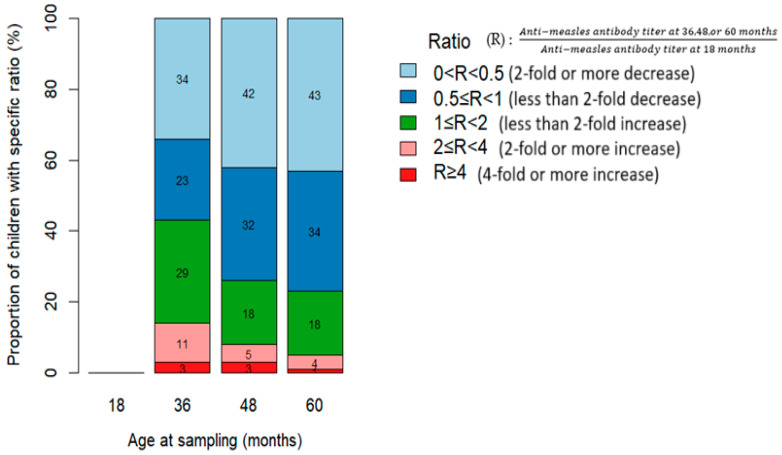
Relative change in anti-measles antibody titers at various time points compared with baseline levels at 18 months among children who received the measles-containing vaccine at either 9 and 15 months or at 6, 9, and 15 months.

**Table 1 vaccines-13-00584-t001:** Sociodemographic characteristics of children and family according to HIV status at inclusion in the ANRS 12225—Pediacam III cohort study, Cameroon, 2007–2013.

N = 496	HIV-Uninfected Infants Born to				HIV-Infected Infant Followed						
	HIV-Infected Mothers n = 180		HIV-Uninfected Mothers n = 173		From the First Week of Life n = 51		From Diagnosis < 7 Months n = 92		*p*-Value1	*p*-Value2	*p*-Value3
	n	%	n	%	n	%	n	%			
Age (months) At inclusion (n = 496), mean [SD]	4.4	[0.9]	4.7	[2.4]	4.4	[4.6]	4.4	[1.4]	0.024	0.001	<0.001
Gender (n = 496)									0.641	0.347	0.601
Female	96	53.3	87	50.3	31	60.8	47	51.1			
Site (n = 496)									0.226	0.700	0.705
Mother and Child Center, Yaoundé	86	47.8	82	47.4	22	43.1	46	50.0			
Laquintinie Hospital, Douala	48	20.2	35	20.2	13	25.5	19	20.7			
Essos Hospital Center, Yaoundé	46	32.4	56	32.4	16	31.4	27	29.3			
Mother’s marital status (n = 494)									0.940	0.793	0.824
Married/cohabiting	119	66.1	116	67.1	32	62.7	52	57.8			
Father’s level of education (n = 378)									0.072	0.968	<0.001
Higher	66	47.1	82	58.6	9	27.3	16	24.6			
Mothers level of Education (n = 492)									<0.001	0.198	<0.001
Higher	39	21.8	79	45.9	6	11.8	4	4.0			
Nutritional status (n = 495)									0.427	0.008	<0.001
WAZ score < −2	5	2.8	2	1.2	7	13.7	33	35.9			
Prematurity (n = 496)									>0.999	0.874	0.365
Yes	21	11.7	21	12.1	10	19.6	13	14.1			
Number of children per home (n = 486)									0.237	>0.999	0.509
Three or more	83	46.9	92	53.8	27	54.0	48	54.5			
Mothers’ profession (n = 492)									0.288	0.179	0.159
Unsalaried	96	53.9	104	60.1	28	54.9	61	67.8			
Functional refrigerator at home (n = 484)									0.009	0.217	<0.001
Yes	96	54.2	115	68.5	27	54.0	37	41.6			
Hemoglobin (g/dL) (n = 485)									0.237	0.733	<0.001
<10	22	12.4	29	17.5	27	54.0	45	49.5			
WHO clinical staging (n = 143)										<0.001	
Stages 3 & 4					5	9.8	38	41.3			
%CD4 lymphocytes At inclusion (n = 143), category											
<25%					23	45.1	60	65.2		0.030	
At inclusion (n = 143), mean [SD]					28.0	[12.5]	21.9	[12.5]		0.002	
Around first MCV (n = 137), mean [SD]					33.3	[11.8]	26.2	[14.4]		0.002	
At 12 months of age (n = 132), mean [SD]					35.5	[10.8]	32.3	[14.1]		0.142	
Viral load											
At inclusion (n = 143), median (log copies/mL) [IQR]					6.3	[5.9–6.9]	6.5	[6.0–7.0]		0.112	
Around first MCV (n = 139) ≥1000 copies/mL					18	38.3	48	52.2		0.170	
At 12 months of age (n = 140) ≥1000 copies/mL					12	25.0	32	34.8		0.321	

*p*-value1: comparison of HIV-uninfected infants groups; *p*-value2: comparison of HIV-infected infant groups; *p*-value3: comparison of four infant groups. SD: standard deviation. MCV: measles-containing vaccine.

**Table 2 vaccines-13-00584-t002:** Measles antibody seropositivity before measles vaccination according to children’s HIV status, ANRS 12225-Pediacam III cohort, 2007–2017, Cameroon.

Pre-Vaccination Measles Antibody Seropositivity at Different Timepoint	Total (n)	HIV-Infected (HI) (n = 153)		HIV-Exposed Uninfected (HEU) (n = 156)		HIV-Unexposed Uninfected (HUU) (n = 147)	
		n (%)	GMT (95%CI)	n (%)	GMT (95%CI)	n (%)	GMT (95%CI)
Before 6 months	156	61 (100)		55 (100)		40 (100)	
Negative	130	58 (95.1)		45 (81.8)		27 (67.5)	
Positive ^‡‡ ǁǁ^	15	1 (1.6)	589	8 (14.5)	642 (450, 890)	6 (15.0)	1041 (571, 1976)
Indetermined	11	2 (3.3)		2 (3.6)		7 (17.5)	
Between 6 and 9 months	275	74 (100)		98 (100)		103 (100)	
Negative	256	69 (93.2)		93 (94.9)		94 (91.3)	
Positive ^ǁǁ $$^	12	3 (4.1)	2208 (546, 6603)	2 (2.0)	532 (423, 669)	7 (6.8)	542 (402, 715)
Indetermined	7	2 (2.7)		3 (3.1)		2 (1.9)	
Between 9 and 12 months	25	18 (100)		3 (100)		4 (100)	
Negative	24	17 (94.4)		3 (100)		4 (100)	
Positive	1	1 (5.6)	377	0 (0.0)		0 (0.0)	
Indetermined	0	0 (0.0)		0 (0.0)		0 (0.0)	
Global ^(a)^	456	153 (100)		156 (100)		147 (100)	
Negative	410	144 (94.1)		141 (90.4)		125 (85.0)	
Positive ^ǁǁ ǁǁ^	28	5 (3.3)	1190 (439, 2805)	10 (6.4)	618 (459, 803)	13 (8.9)	732 (503, 1045)
Indeterminate	18	4 (2.6)		5 (3.2)		9 (6.1)	
Age (months) at testing, median (IQR) ^‡‡^		6.3 (5.3, 7.8)		6.0 (5.6, 6.3)		6.1 (5.9, 6.4)	

^‡‡^ 0.01 ≤ *p* < 0.05 ^$$^ 0.05 ≤ *p* ≤ 0.10 ^ǁǁ^ *p* > 0.10. ^Symbol ^^color-red^: Chi-square test for equality of proportions of subjects with measles antibody among different groups. ^Symbol ^^color-blue^: Kruskal–Wallis’s rank sum test for the equality of geometric mean titer among different groups. ^(a)^: Considering « Indeterminate » as seropositive, chi-square test for equality of proportions is significant at 5% (HI: 9 (5.9%), HEU: 15 (9.6%), HUU: 21 (15.0%), *p* = 0.032) and significant before 6 months (*p* = 0.025). GMT: Geometric mean titers.

**Table 3 vaccines-13-00584-t003:** Measles antibody seropositivity after measles vaccination at different time points and according to children’s HIV status, ANRS 12225—Pediacam III, 2007–2017, Cameroon.

Measles Containing Vaccine (MCV) Administered at	Tested for Measles Antibody	HIV-Exposed Uninfected (n = 180)		HIV-Unexposed Uninfected (n = 173)		HIV-Infected (n = 143)	
		n (%)	GMC (95%CI)	n (%)	GMC (95%CI)	n (%)	GMC (95%CI)
6 months MCV (184)	60 (32.6%)	12				48	
Negative ^ǁǁ^	14	4 (33.3)		-		10 (20.8)	
Positive ^ǁǁ ǁǁ^	43	8 (66.7)	668 (469, 878)	-		35 (72.9)	1116 (835, 1463)
Indeterminate ^ǁǁ^	3	0 (0.0)		-		3 (6.2)	
9 months MCV (302)	234 (77.5%)	45		131		58	
Negative ^ǁǁ^	33	7 (15.6)		21 (16.0)		5 (8.6)	
Positive ^ǁǁ, ǁǁ^	176	34 (75.6)	1175 (818, 1618)	96 (73.3)	1053 (883, 1274)	46 (79.3)	893 (701, 1082)
Indeterminate ^ǁǁ^	25	4 (8.8)		14 (10.7)		7 (12.1)	
6 + 9 months MCV (152)	126 (82.9%)	80				46	
Negative ^ǁǁ^	11	6 (7.5)		-		5 (10.9)	
Positive ^ǁǁ, ǁǁ^	107	69 (86.3)	1126 (971, 1328)	-		38 (82.6)	1068 (789, 1416)
Indeterminate ^ǁǁ^	8	5 (6.2)		-		3 (6.5)	
6 + 15 months MCV (26)	21 (80.8%)	10				11	
Negative	0	0 (0.0)		-		0 (0.0)	
Positive ^ǁǁ, ǁǁ^	21	10 (100)	1781 (1039, 2934)	-		11 (100.0)	1626 (970, 2901)
Indeterminate	0	0 (0.0)		-		0 (0.0)	
9 + 15 months MCV (238)	170 (71.4%)	33		111		26	
Negative	0	0 (0.0)		0 (0.0)		0 (0.0)	
Positive ^‡‡, ǁǁ^	167	31 (93.9)	1990 (1547, 2549)	111 (100.0)	1878 (1640, 2169)	25 (96.2)	1665 (1177, 2348)
Indeterminate ^‡‡^	3	2 (6.1)		0 (0.0)		1 (3.8)	
6 + 9 + 15 months MCV(125)	95 (76.0%)	60				35	
Negative	0	0 (0.0)		-		0 (0.0)	
Positive ^ǁǁ, $$^	93	59 (98.3)	1617 (1307, 1992)	-		34 (97.1)	2171 (1695, 2796)
Indeterminate ^ǁǁ^	2	1 (1.7)		-		1 (2.9)	

^‡‡^ 0.01 ≤ *p* < 0.05 ^$$^ 0.05 ≤ *p* ≤ 0.10 ^ǁǁ^
*p* > 0.10. ^Symbol ^^color–red^: Chi-square test for equality of proportions of subject with measles antibody among the different groups. ^Symbol ^^color–blue^: Kruskal–Wallis’s rank sum test for the equality of geometric mean titer among the different groups.

## Data Availability

The raw data supporting the results and conclusions of this article will be made available by the authors upon request, with consideration of the participants’ privacy and ethics.

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
