# Peer review of "Pre- and Post-Vaccination Measles Antibody and Persistence Up to 5 Years of Age Among Early ART-Treated HIV-Infected, HIV-Exposed Uninfected and HIV-Unexposed Children in Cameroon"

_vaccines, 2025, doi:10.3390/vaccines13060584_

Round 1

Reviewer 1 Report

Comments and Suggestions for Authors

The authors in this manuscript discuss an important and novel topic which is the comparison between levels of antibodies of measles before and 5 years after measles vaccination in children who had HIV and receive ART, children exposed to HIV but uninfected and children unexposed uninfected with HIV. The manuscript is well designed and written. Some comments are to be addressed:

  1. There are too many and too long keywords. 
  2. In introduction, the authors need to explain the problem of both HIV and measles in Cameroon in more details.
  3. In the HIV infected children, what was their HIV viral load?
  4. Did the mothers of infected and the exposed children receive ART or not? and did they achieve viral suppression? please, add these details.
  5. In the tables, determine the age, was it in months or years?
  6. In table 2, explain the abbreviation of GMT in the footnote.
  7. In table 1,2,3, the results of comparisons are not clear and confusing. Using symbols to indicate p value makes the interpretation difficult. Kindly, explain the comparison in a more clear and easy way.
  8. In discussion, line 305 explain the abbreviation of HAART before the word.

Author Response

On behalf of my co-authors, I deeply appreciate the comments and suggestions from you and the reviewers, which have greatly improved our manuscript.

Comments and Suggestions for Authors

The authors in this manuscript discuss an important and novel topic which is the comparison between levels of antibodies of measles before and 5 years after measles vaccination in children who had HIV and receive ART, children exposed to HIV but uninfected and children unexposed uninfected with HIV. The manuscript is well designed and written. Some comments are to be addressed:

  1. There are too many and too long keywords

Thanks for the advice. We have modified and removed some of the keywords, reducing the number from 10 to 8 as presented below:

"Measles humoral immunity; Persistence of measles antibodies; pre-measles vaccine antibodies; early treated HIV-infected; HIV-exposed uninfected; HIV-unexposed uninfected; low- and middle-income countries; real life conditions"

2. In introduction, the authors need to explain the problem of both HIV and measles in Cameroon in more details

Thank you for the remark. The short paragrapgh below has been added in the introduction (lines 57-64).

“The second dose of the measles vaccine, targeting children aged 15 months, was introduced in 2020, with an estimated uptake of 45% in 2023. In the same year, the country reported 6,105 measles cases, an increase of approximately 90% compared to 2022. According to the most recent Demographic and Health Survey (DHS) conducted in 2018 in Cameroon, HIV prevalence among adults aged 15–49 years declined from 5.4% in 2004 to 4.3% in 2011, and further to 2.7% in 2018. The HIV prevalence was higher among women (3.4%) than men (1.9%)”

3. In the HIV infected children, what was their HIV viral load?

In Table 1, we added data on HIV viral load at inclusion, prior to measles vaccine administration, and at approximately 12 months of age

4. Did the mothers of infected and the exposed children receive ART or not? and did they achieve viral suppression? please, add these details.

Some mothers of HIV-infected or HIV-exposed uninfected children did receive antiretroviral therapy (ART). However, HIV viral load was not routinely measured at the time, as it was not included in the standard panel of biological tests offered free of charge under national guidelines. Only a limited number of pregnant women who could afford the test had it performed. As these data were not collected in our study, we are unfortunately unable to provide further details.

I would like to recall the inclusion process of children in the PEDIACAM study. Infants born to HIV-infected mothers were recruited at the maternity wards or pediatric services of participating sites during the first postnatal week, between November 2007 and October 2011. HIV-infected infants who presented after the first postnatal week and were diagnosed before the age of 7 months were also enrolled.

In the group of HIV-infected infants, fewer than half of the mothers received treatment, including combination antiretroviral therapy (cART) initiated before conception, during pregnancy, or as a short-course regimen. In the group of HIV-exposed but uninfected infants, fewer than 5% of mothers did not receive any antiretroviral therapy for the prevention of mother-to-child transmission (PMTCT).

5. In the tables, determine the age, was it in months or years?

Age is expressed in months and has been added to Tables 1 and 2

6. In table 2, explain the abbreviation of GMT in the footnote

The explanation of the abbreviation GMT (Geometric Mean Titers) has been added to the footnotes of Table 2

7. In table 1,2,3, the results of comparisons are not clear and confusing. Using symbols to indicate p value makes the interpretation difficult. Kindly, explain the comparison in a more clear and easy way

It was not our intention to cause confusion. In this presentation, our goal was to simplify the tables. Given the substantial amount of data, certain information was placed in the table footnotes. P-values are indicated using symbols or special characters, along with a color code specifying the type of statistical test applied. This explanation has now been included in the footnotes, with the expectation that it will facilitate interpretation.

8. In discussion, line 305 explain the abbreviation of HAART before the word

Thank you for the remark. The abbreviation HAART (Highly Active Antiretroviral Therapy) has been explained

Reviewer 2 Report

Comments and Suggestions for Authors
  1. The study presents a valuable examination of measles antibody persistence in HIV-infected, HIV-exposed uninfected (HEU), and HIV-unexposed uninfected (HUU) children. While previous studies have explored measles vaccination efficacy in HIV-exposed populations, this research provides a nuanced longitudinal perspective, spanning up to five years, which enhances its originality. The integration of real-world vaccination schedules, antibody decay kinetics, and external validation through comparative studies strengthens its contribution to the field. However, greater emphasis on novel immunological insights, such as T-cell responses, could further distinguish it from existing literature.

  1. The article is generally well-structured, with a logical flow from methodology to results and discussion. However, some sections could benefit from greater precision in reporting statistical findings, for instance, outlining confidence intervals alongside p-values to strengthen numerical interpretations. Additionally, while the discussion references global studies for comparison, these contrasts could be more clearly delineated to avoid ambiguity. The readability is strong, but refining complex immunological concepts for broader accessibility could make the findings more impactful.

  1. The argument progresses systematically, with a clear connection between vaccination schedules, immune response, and waning antibody levels over time. The findings are well-supported by comparative analyses with external studies, reinforcing their credibility. However, greater exploration of causal relationships, particularly regarding maternal antibody transfer inefficiencies and HIV-driven immune alterations, would deepen the discussion. Some sections such as the impact of immunization campaigns on antibody fluctuations could also be more explicitly tied to the study’s primary objectives to improve overall coherence.

  1. Some methodological refinements such as analyzing alternative immune markers beyond antibodies or discussing the feasibility of implementing vaccination schedules in real-world HIV care settings could further enhance its contribution. Additionally, addressing logistical barriers to vaccination adherence in resource-limited environments would strengthen its global applicability.

Suggestions for Improvement

  1. Expand Immunological Context: Incorporating cell-mediated immunity findings (T-cell function) alongside antibody trends would improve depth.
  2. Clarify Statistical Reporting: Include confidence intervals with p-values for improved transparency.
  3. Refine Global Comparisons: Ensure comparative studies are distinctly contextualized to avoid ambiguity.
  4. Explore Vaccination Policy Implications: Discuss how findings could inform national immunization strategies for HIV-exposed children.
  5. Examine Long-Term Immunity Trends: Extend the discussion on how waning immunity might necessitate additional booster doses.

Author Response

Comments and Suggestions for Authors

The study presents a valuable examination of measles antibody persistence in HIV-infected, HIV-exposed uninfected (HEU), and HIV-unexposed uninfected (HUU) children. While previous studies have explored measles vaccination efficacy in HIV-exposed populations, this research provides a nuanced longitudinal perspective, spanning up to five years, which enhances its originality. The integration of real-world vaccination schedules, antibody decay kinetics, and external validation through comparative studies strengthens its contribution to the field. However, greater emphasis on novel immunological insights, such as T-cell responses, could further distinguish it from existing literature.

The article is generally well-structured, with a logical flow from methodology to results and discussion. However, some sections could benefit from greater precision in reporting statistical findings, for instance, outlining confidence intervals alongside p-values to strengthen numerical interpretations. Additionally, while the discussion references global studies for comparison, these contrasts could be more clearly delineated to avoid ambiguity. The readability is strong, but refining complex immunological concepts for broader accessibility could make the findings more impactful.

The argument progresses systematically, with a clear connection between vaccination schedules, immune response, and waning antibody levels over time. The findings are well-supported by comparative analyses with external studies, reinforcing their credibility. However, greater exploration of causal relationships, particularly regarding maternal antibody transfer inefficiencies and HIV-driven immune alterations, would deepen the discussion. Some sections such as the impact of immunization campaigns on antibody fluctuations could also be more explicitly tied to the study’s primary objectives to improve overall coherence.

Some methodological refinements such as analyzing alternative immune markers beyond antibodies or discussing the feasibility of implementing vaccination schedules in real-world HIV care settings could further enhance its contribution. Additionally, addressing logistical barriers to vaccination adherence in resource-limited environments would strengthen its global applicability.

Suggestions for Improvement

Thank you to the Reviewer for the appreciation and suggestions 

  1. Expand Immunological Context: Incorporating cell-mediated immunity findings (T-cell function) alongside antibody trends would improve depth.

We would have liked to include aspects related to the cellular immune response to provide a more comprehensive and holistic view. However, challenges related to cell preservation prevented us from achieving this objective. That said, as PEDIACAM is an ongoing cohort study, we are planning to assess long-term (after 10 years of age) vaccine-induced immunity, integrating both humoral and cell-mediated responses.

2. Clarify Statistical Reporting: Include confidence intervals with p-values for improved transparency.

Thanks for this remark. We try to clarify it.

For some findings, we added confidence intervals (line 190; lines 255-256) to complement those already provided (line 201, Table 1, 2, and 3)

Given the substantial amount of data, certain information was placed in the table footnotes. P-values are indicated using symbols or special characters, along with a color code specifying the type of statistical test applied. This explanation has now been included in the footnotes (Tables 2 & 3), with the expectation that it will facilitate interpretation

3. Refine Global Comparisons: Ensure comparative studies are distinctly contextualized to avoid ambiguity

We have strived to make throughout the discussion of our results

4. Explore Vaccination Policy Implications: Discuss how findings could inform national immunization strategies for HIV-exposed children.

The results from our study on HIV-exposed uninfected (HEU) children are generally reassuring in terms of their ability to tolerate and respond to the administered vaccines. However, it is important to highlight that the measles vaccine was administered to a significant number of children starting at six months of age, a strategy that our program faces challenges in implementing, primarily due to the potential stigma it could cause, unless appropriate measures are taken beforehand (lines 367-374)

5. Examine Long-Term Immunity Trends: Extend the discussion on how waning immunity might necessitate additional booster doses.

Thank you for this suggestion. In our discussion, we highlighted the need to consider this approach, particularly in light of the continued widespread circulation of wild-type measles virus in our setting (lines 379-380)

Reviewer 3 Report

Comments and Suggestions for Authors

This paper reports the strong antibody response to measles of HIV-infected infants by comparison to two groups of uninfected children. The measure of antibodies before the first immunization is also a strength of this prospective study. The sustained response through time during the 5 following years in the three groups of tested subjects is an additional encouraging feature of the study. The discussion proposes an interesting overview of concurrent studies that analyzed similar populations within the African context. The absence of prospective clinical follow-up documenting possible immune failure could be acknowledged as a limit of the study.    

I support the publication of the paper. However, a few minor points could be improved:

  • The title could be shortened; is it necessary to identify at this stage the 3 groups of tested patients?
  • Line114: please precise the meaning of the sentence: “Since it is currently available at a cost, the project provided this dose”.
  • Figure 2: Please use the same scale for ordinates for the three panels.
  • Figure 5: At 36 months, the sum of the five groups is 101 and not 100.

The English style could need to be simplified to improve the understanding of the text.

Comments on the Quality of English Language

I suggest a simplification of the English language to improve the understanding of the text.

Author Response

Comments and Suggestions for Authors

This paper reports the strong antibody response to measles of HIV-infected infants by comparison to two groups of uninfected children. The measure of antibodies before the first immunization is also a strength of this prospective study. The sustained response through time during the 5 following years in the three groups of tested subjects is an additional encouraging feature of the study. The discussion proposes an interesting overview of concurrent studies that analyzed similar populations within the African context. The absence of prospective clinical follow-up documenting possible immune failure could be acknowledged as a limit of the study.    

I support the publication of the paper. However, a few minor points could be improved:

We thank the Reviewer 3 for the overall appreciation and positive comment of this work. We would like to address your comment regarding the 'absence of prospective clinical follow-up of the children'. The Pediacam cohort remains active to date, and the children continue to be followed regularly. The fact that these children live in communities where measles cases are occasionally reported, yet no significant number of clinical cases has been observed among cohort participants, may be interpreted as an indication of the effectiveness of vaccination.

1. The title could be shortened; is it necessary to identify at this stage the 3 groups of tested patients?

We thank you for this suggestion. However, we considered it appropriate to retain the mention of the different groups in the title to ensure greater clarity and to highlight the specific features of our study

2. Line114: please precise the meaning of the sentence: “Since it is currently available at a cost, the project provided this dose”.

We simply wanted to clarify that this vaccine dose, administered at 15 months, was provided free of charge to all children by the project.

The sentence now read as below

“A catch-up dose of the measles vaccine, administered as the Measles-Mumps-Rubella (MMR, Sanofi Pasteur) vaccine, is scheduled at 15 months (MCV2). Given that this dose is currently available at a cost, the project has provided it to the beneficiaries.” (lines 119-122)

3. Figure 2: Please use the same scale for ordinates for the three panels.

Thank you for your remark. We have made the correction using the same scale to ensure consistency. Additionally, we have revised the Y-axis title from 'Percentage of seropositive (%)' to 'Seropositive to measles antibodies (%)' for greater clarity." (page 10)

4. Figure 5: At 36 months, the sum of the five groups is 101 and not 100

Thank you for this observation. This error was due to rounding. It has been corrected (page 12)

5. The English style could need to be simplified to improve the understanding of the text.

Thank you for the remark. A senior researcher, who is a native English speaker, reviewed again the manuscript and made some revisions, particularly in keywords and in the text (lines 322–324 and 328–330)

Reviewer 4 Report

Comments and Suggestions for Authors

The authors conducted a study of measles antibody titers during the first five years of life in HIV-infected, HIV-exposed uninfected, and HIV-unexposed uninfected children in Cameroon. Most of the figures in this manuscript are not sufficiently informative and require revision.

Some comments are listed below:
1. Line 64. Please provide data on the prevalence of HIV infection among children in Cameroon, including children under 5 years of age.
2. Line 79. It should be clarified here that the group of HIV-exposed uninfected children are children born to HIV-positive mothers.
3. Line 90. Do you have a link to this study? A link is required.
4. Figure 2. Please provide a boxplot with dots representing antibody titers in mUI/mL. The % of seropositive children with a confidence interval can be provided above the graph.
5. Figure 3. Please plot the data points on the graph. The standard deviation should be reported for each data point. Why is a logarithmic scale used here? Please report the antibody titers in mUI/mL.
6. Figures 4 and 5. This is not clear. What is the relative change in titer? It is not clear what 100% means in Figure 5. These graphs should be replaced. The titer change should be plotted as a line graph with time points.

Author Response

Comments and Suggestions for Authors

The authors conducted a study of measles antibody titers during the first five years of life in HIV-infected, HIV-exposed uninfected, and HIV-unexposed uninfected children in Cameroon. Most of the figures in this manuscript are not sufficiently informative and require revision.

Some comments are listed below:
1. Line 64. Please provide data on the prevalence of HIV infection among children in Cameroon, including children under 5 years of age.

Reliable national data on HIV prevalence among children under 15 in Cameroon is lacking due to their exclusion from the Demographic and Health Survey. In 2023, an estimated 27,960 children under 15 were living with HIV, representing 5.7% of the 490,484 people living with HIV. There were 3,657 new pediatric HIV infections. Among 12,957 HIV-exposed infants, 96.8% received early PCR testing, with a positivity rate of 3.3%. The overall mother-to-child transmission rate, including breastfeeding, was estimated at 15% at 12 months. (lines 63 - 68)

  1. Line 79. It should be clarified here that the group of HIV-exposed uninfected children are children born to HIV-positive mothers

This has been clarified by adding “born to HIV-infected mothers”, and the sentence now reads:
To address some of these gaps, we analyzed data from the ANRS 12225-Pediacam III cohort, which included early-treated HIV-infected (HI) children followed from birth or diagnosis (before 7 months of age), alongside two uninfected control groups: HIV-exposed uninfected (HEU) children born to HIV-infected mothers. Line 86

  1. Line 90. Do you have a link to this study? A link is required.

While there isn't a dedicated website for the ANRS 12225-Pediacam III study, you can find pertinent information through the institutions involved:

  • Centre Pasteur du Cameroun (CPC): As the coordinating center for the study, CPC provides details about its research activities, including Pediacam, on its official website (https://www.pasteur-yaounde.org/index.php/fr/nos-recherches-vih-sida/projets-pediacam-i-et-ii).
  • ANRS | Maladies Infectieuses Émergentes (ANRS MIE): The French National Agency for Research on AIDS and Viral Hepatitis, which funds the study, offers information on its research programs and collaborations

   4. Figure 2. Please provide a boxplot with dots representing antibody titers in mUI/mL. The % of seropositive children with a confidence interval can be provided above the graph.

We modified Figure 2 as requested by providing a boxplot with dots and the percentage of seropositive to measles antibodies with confidence interval 

  1. Figure 3. Please plot the data points on the graph. The standard deviation should be reported for each data point. Why is a logarithmic scale used here? Please report the antibody titers in mUI/mL.

We have modified Figure 3 as suggested.
Regarding the use of a logarithmic scale, we considered it appropriate due to the wide range of antibody titers observed in this study (from 23.8 to 22,735.6 mUI/mL), spanning several orders of magnitude. A logarithmic scale is particularly suitable in this context, as it facilitates visualization across the full range of values and highlights relative (i.e., percentage) changes more effectively.

  1. Figures 4 and 5. This is not clear. What is the relative change in titer? It is not clear what 100% means in Figure 5. These graphs should be replaced. The titer change should be plotted as a line graph with time points.

We clarified the presentation of Figures 4 and 5 by adding formulas to define the relative difference (Figure 4) and the relative change (Figure 5).

In Figure 4,

Relative difference =(antimeasles antibody titer at 36, 48, or 60 months - antimeasles antibody titer at 18 months)*100/antimeasles antibody titer at 18 months   

in Figure 5,

Ratio (R) =antimeasles antibody titer at 36, 48, or 60 months/antimeasles antibody titer at 18 months

These figures present a point-by-point quantification of the evolution of antibody titers from 18 months (used as the baseline) to 60 months of follow-up.
In Figure 5, we removed the bar at 18 months, as it could be misleading. It was initially included to indicate that 100% of participants had a ratio of 1 at 18 months, since the measles antibody titer at that time point, divided by itself, equals one.
We hope these clarifications enhance the reader's understanding and support our decision not to replace the figures.

Round 2

Reviewer 4 Report

Comments and Suggestions for Authors

The new additions to the manuscript made a big difference. The quality of the paper had improved, and all my questions were addressed. No more comments.